# The Relationship between Environmental Regulation and Green Total Factor Productivity in China: An Empirical Study Based on the Panel Data of 177 Cities

**DOI:** 10.3390/ijerph17155287

**Published:** 2020-07-22

**Authors:** Mingliang Zhao, Fangyi Liu, Wei Sun, Xin Tao

**Affiliations:** 1Department of International Economics and Trade, Shandong University of Finance and Economics, Jinan 250002, China; mlzhao@sdufe.edu.cn (M.Z.); fangyil@mail.sdufe.edu.cn (F.L.); 2Key Laboratory of Regional Sustainable Development Modeling, Institute of Geographic Sciences and Natural Resources Research, Chinese Academy of Sciences, Beijing 100101, China; 3College of Resources and Environment, University of Chinese Academy of Sciences, Beijing 100049, China; 4Department of Geography, University at Buffalo, the State University of New York, Buffalo, NY 14261, USA; xintao@buffalo.edu

**Keywords:** industrialization, environmental regulation, green total factor productivity, systematic GMM

## Abstract

Promoting the coordinated development of industrialization and the environment is a goal pursued by all of the countries of the world. Strengthening environmental regulation (ER) and improving green total factor productivity (GTFP) are important means to achieving this goal. However, the relationship between ER and GTFP has been debated in the academic circles, which reflects the complexity of this issue. This paper empirically tested the relationship between ER and GTFP in China by using panel data and a systematic Gaussian Mixed Model (GMM) of 177 cities at the prefecture level. The research shows that the relationship between ER and GTFP is complex, which is reflected in the differences and nonlinearity between cities with different monitoring levels and different economic development levels. (1) The relationship between ER and GTFP is linear and non-linear in different urban groups. A positive linear relationship was found in the urban group with high economic development level, while a U-shaped nonlinear relationship was found in other urban groups. (2) There are differences in the inflection point value and the variable mean of ER in different urban groups, which have different promoting effects on GTFP. In key monitoring cities and low economic development level cities, the mean value of ER had not passed the inflection point, and ER was negatively correlated with GTFP. The mean values of ER variables in the whole sample, the non-key monitoring and the middle economic development level cities had all passed the inflection point, which gradually promoted the improvement of GTFP. (3) Among the control variables of the different city groups, science and technology input and the financial development level mainly had positive effects on GTFP, while foreign direct investment (FDI) and fixed asset investment variables mainly had negative effects.

## 1. Introduction

Since the beginning of the 21st century, China has made great progress in environmental protection. However, with the progress of industrialization, the contradiction between economic growth and environmental protection has become more prominent. In 2000, China’s industrial added value reached 4.03 trillion yuan. By 2019, it had jumped to 31.71 trillion yuan, an increase of 6.9 times [1]. China has also become the ‘world’s factory’. However, China is still mainly concentrated in the middle and low-end links of the global value chain’s division of labor, and characteristics of ‘high consumption’, ‘low technology’ and ‘high emissions’ are still relatively obvious in the process of industrialization. Such a division of labor patterns and production characteristics results in intensified environmental pollution, such as the Songhua River water pollution incident in 2005 and the blue-green algae pollution incident in Taihu Lake in 2007. Therefore, the Chinese government has strengthened environmental protection. On the one hand, it has increased investment in environmental protection and improved its environmental governance capacity. On the other hand, it has introduced more stringent environmental protection policies to strengthen environmental regulation (ER). Nevertheless, can strengthening ER promote environmental improvement and green total factor productivity (GTFP)? Around this issue, domestic and foreign scholars have carried out extensive and in-depth research, but the conclusions reached are inconsistent.

This paper takes 177 cities at the prefecture level and above in China as its object, and empirically studies the correlation between ER and GTFP in China’s industrialization process by using urban panel data and a systematic Gaussian Mixed Model (GMM) model from 2007 to 2016. In addition, this paper also analyzes the non-linear relationship between ER and GTFP, and the difference of this relationship between different cities. The main contribution of this article consists of the following aspects: firstly, the existing studies mostly focus on the relationship between ER and GTFP from a static perspective, ignoring the possible dynamic inertia in the process of industrialization. We introduce the dependent variable lag term into the model for regression analysis in order to avoid the possible deviation in the model estimation. Secondly, most existing studies believe that there is a linear relationship between ER and GTFP. After introducing the square term of ER, we found that there is a nonlinear relationship between them, and calculated the inflection points respectively. Thirdly, most of the existing studies are based on different industries, national and provincial administrative units, and few examples in the literature are based on prefecture-level cities. Moreover, few studies have been conducted on the differences among different cities according to their regulation intensity and development level. We divided the study objects into key monitoring cities and non-key monitoring cities, as well as cities with a high economic development level, a medium economic development level and a low economic development level. In order to highlight the pertinence of the suggestions, we studied the differences of this relationship between different cities. At the same time, we believe that economic development and environmental pollution are accompanied by each other. Especially in the process of rapid industrialization, environmental pollution caused by economic development may be inevitable. Therefore, the research conclusions and suggestions may have significance as references for developing countries in the process of rapid industrialization.

The structure of this paper is as follows: the second part is literature review. The third part is the econometric model, variables and data. The fourth part is the empirical results and analysis. The fifth part is the conclusion and policy suggestions.

## 2. Literature Review

### 2.1. Theoretical Background

The first theory is the environmental Kuznets Curve Theory. This theory points out that there is an inverted U-shaped relationship between economic development and environmental pollution. When a country’s economic development level is low, environmental pollution will gradually worsen with economic development and the increase of per capita income. When economic development reaches a certain level, or a certain critical point or ‘inflection point’, environmental pollution will gradually slow down and the environmental quality will be improved as economic development and per capita income increase. This phenomenon is known as the environmental Kuznets Curve [2]. After the environmental Kuznets Curve was proposed, scholars tried to explain the environmental Kuznets Curve theoretically from the aspects of scale effect, technical effect, structural effect, environmental quality demand and ER. For example, Banister and Berechma found that economic development leads to improved transportation accessibility, and may potentially trigger several major positive externalities that increase productivity, reduce production costs and promote the more efficient use of resources [3]. The environmental Kuznets Curve theory inspires us to think whether there is a similar relationship between ER and GTFP.

The second theory is the Porter hypothesis. Proper ER can promote enterprises to carry out more innovative activities, which will improve the productivity of enterprises, offset the cost brought by environmental protection, improve the profitability of enterprises in the market, and improve the quality of products, which may enable domestic enterprises to gain competitive advantages in the international market [4]. Prior to porter’s hypothesis, it was widely believed that ER was the main factor contributing to the increase in enterprise costs, and that it would have a negative impact on productivity and competitiveness (Figure 1). Therefore, the key issue is whether ER promotes enterprise innovation and further improves production efficiency.

The third theory is the pollution heaven hypothesis, which mainly refers to the tendency of enterprises in pollution-intensive industries to be established in countries or regions with relatively low environmental standards. When there is a uniform price for the product, the cost of production determines the location of production. If all countries have the same conditions except for environmental standards, then polluting enterprises will choose to produce in countries with lower environmental standards, which will become a paradise for pollution [5].

### 2.2. Empirical Literature

The influence of ER on GTFP is not a new topic, which could be said to be a long-standing problem. Since the 1990s, as sustainable development has been widely recognized by the international community, more and more attention has been paid to the three dimensions of economic, environmental and social development. How to realize a ‘win–win’ between economic growth and environmental protection has become a hot topic in academic research. In this context, ER is undoubtedly one of the important measures to achieve the goal of a ‘win–win’. According to the literature retrieval, the existing researches mainly focus on two aspects. The first is industry-specific research. The second aspect is location-specific research.

In terms of industry-specific research, Liang et al. [6] concluded that ER had a significant positive effect on the growth of the GTFP of the logistics industry in Jiangsu Province, China, in the years 2006–2018, and that technological progress has become an important endogenous force that promotes the GTFP of the logistics industry. Weber and Domazlicky [7] constructed a Malmquist–Luenberger index of total factor productivity growth for manufacturing, and accounted for the toxic emissions in manufacturing during 1988–1994. The foundings suggest that the failure to account for toxic emissions in manufacturing results in a significant understatement of TFP growth. Managi et al. [8] applied Data Envelopment Analysis techniques to measure changes in TFP in the oil and gas production in the Gulf of Mexico’s outer continental shelf. The results show the direction of causality between technological innovation and ER, and found support for a recast version of the Porter hypothesis. Lanoie et al. [9] provide an empirical analysis of the relationship between the stringency of ER and TFP growth in the manufacturing sector of Quebec. The results suggest that the contemporaneous impact of ER on productivity is negative, and that the Porter hypothesis is more relevant for more polluting sectors and sectors that are more exposed to international competition. Hamamoto [10] took Japanese manufacturing industries as examples, and found that increases in R&D investment stimulated by the regulatory stringency have a significant positive effect on the growth rate of TFP. Garrett et al. [11] used the theory of agglomeration economies to understand the development of soybean production in two counties along the Santarém–Cuiaba (BR-163) highway in the Brazilian Amazon: Santarém, Pará and Sorriso, Mato Grosso. They found that the supply chain became extremely competitive and diverse in Sorriso, where few ER existed, while environmental restrictions reduced the diversification of the supply chain in Santarém. The presence of a soy agglomeration economy in Sorriso spurred innovation, increased productivity, and led to extremely rapid soy expansion in that county. Berman and Bui [12] examined the effect of air quality regulation on the productivity of some of the most heavily regulated manufacturing plants in the United States, the oil refineries of the Los Angeles Air Basin. They found that, despite the high costs associated with the local regulations, productivity in the Los Angeles Basin refineries rose sharply during the 1987–1992 period. Arocena and Price [13] tested for the effect of ER in reducing pollutants on both publicly and privately owned Spanish electricity generators, and found that public firms are more efficient under the cost of service regulation, and that incentive regulation will increase efficiency in the private sector.

There is less research on specific areas than on specific industries. Du et al. [14] used the General Malmquist–Luenberger productivity index to estimate the industrial green competitiveness in China’s 30 provinces in 2001–2016, and investigated the compound effect of ER and governance transformation on the improvement of China’s industrial green competitiveness. They found that the relationship between ER and industrial green competitiveness presented a U-shaped curve, which verified the Porter hypothesis. Wang et al. [15] established a translog stochastic frontier production function model, and calculated the TFP and its composition in China from 1995 to 2015. They found that ER slows down the growth rate of TFP in China and all of its regions, and that the growth rate of TFP is the fastest in the east of China and the slowest in the central region. Li and Wu [16] employed a metafrontier Malmquist–Luenberger index and a spatial Durbin model to investigate the influence of both local and civil ER and its spatial spillover effect on GTFP. The results show that effect of local ER on GTFP is significantly positive in high political attribute cities, but has negative effects in lower political attribute cities.

To sum up, we found that although some scholars have pointed out that the relationship between ER and GTFP is different in different industries or areas, and there may be a U-shaped relationship between the two, there are few existing research results, and even fewer studies combine the difference and the U-shaped relationship. Therefore, we introduce the concept of GTFP, add the unexpected output, such as air pollutant emission, and construct a dynamic panel model from the city level and the system’s GMM, which makes up for the endogenous problem caused by the use of a static model and improves the robustness of the empirical analysis. At the same time, the possible threshold effect and non-linear relationship between ER and GTFP are investigated in order to clarify the possible reasons for the current controversy.

## 3. Research Methods and Data Sources

This paper applies the environmental monitoring data and economic data of 177 cities above the prefecture level in China from 2007 to 2016 to examine the complex relationship between ER and GTFP. Due to the lag of China’s statistical data, the latest data available is 2018. However, due to the lack of data on pollutant emissions and foreign direct investment (FDI) in some cities in 2017–2018, we chose the data applied to 2016 to ensure the reliability and accuracy of the analysis results. It is important to note that we used long-term panel data with regularity and trends to carry out our research, rather than annual descriptive statistics, which can also ensure that our research results reflect the real relationship between ER and GTFP.

### 3.1. Variable Selection

The variables include explained variables and explanatory variables. Besides ER, four control variables, including FDI, were added as explanatory variables.

#### 3.1.1. Explained Variables

TFP is the combined productivity of production units as total elements in a system, often expressed as the ratio of the total output in a system to the total production factor inputs. GTFP takes the factors of production and output, such as energy inputs and environmental pollution, into account. In this paper, we refer to Färe and Grosskopf’s [17] method to measure GTFP in cities based on the Malmquist-Luenberger (ML) index of non-radial slacks-based measure (SBM) directional distance. The ML index for periods *t* to *t* + 1 is:(1)$$ML_{t}^{t+1}={\left[\frac{1+\vec{D_{0}^{t}}\left(x^{t},y^{t},z^{t},g^{t}\right)}{1+\vec{D_{0}^{t}\left(x^{t+1},y^{t+1},z^{t+1},g^{t+1}\right)}}\times \frac{1+\vec{D_{0}^{t+1}}\left(x^{t},y^{t},z^{t},g^{t}\right)}{1+\vec{D_{0}^{t+1}\left(x^{t+1},y^{t+1},z^{t+1},g^{t+1}\right)}}\right]}^{\frac{1}{2}}$$
where $\vec{D}$ is the directional distance function, $x^{t}$, $y^{t}$ is the input indicator, and $z^{t}$ and $g^{t}$ are the expected and non-expected outputs.

The input indicators include: (1) labor input: the number of people employed; (2) capital input: the capital stock of each city; (3) energy consumption, expressed in terms of urban electricity consumption.

The output indicators include: (1) the expected output, i.e., Gross Domestic Product; (2) unexpected outputs, e.g., sulfur dioxide emissions and dust emissions.

#### 3.1.2. Explanatory Variables and Control Variables

Environmental Regulation (*ER_it_*): according to the literature search, there are five main ways to measure ER [18]. (1) Low and Yeats (1992) [19] used the number of local government decrees to measure the intensity of environmental regulation. (2) The intensity of environmental regulation is measured by the income from sewage charges, such as in Levinson (1996) [20]. (3) Environmental regulation intensity is measured by the proportion of pollution control investment in the total cost of enterprises, such as Berman and Bui (2001) [12]. (4) The intensity of environmental regulation is measured by per capita GDP (income), such as in Mani and Wheeler (1998) [21]. (5) The intensity of environmental regulation is measured by different pollutant emission densities, such as in Cole and Elliott (2003) [22].

In China, there is no fixed mode of government intervention, and there is no independent regulatory tool. The first and second methods can only reflect one aspect of environmental regulation measures, but not the entirety of the environmental regulation measures. The third method mainly considers the pollutant emission control of manufacturing enterprises, and the object of this paper is the city—urban pollutant emission and environmental regulation are not limited to the manufacturing industry. The fourth method ignores the emission of pollutants, which is not accurate. The fifth method involves the change of emission intensity of different pollutants, which is not available in China at present. At the same time, different pollutant emission units are inconsistent, and their intensity changes are generally aggregated. If only one pollutant emission intensity is used to measure, it is not accurate. Referring to the existing research indicators, and considering that this paper takes the city as its research object, it measures the environmental regulation from the perspective of air pollution. This paper improves the method of environmental regulation, measures the intensity of environmental regulation more scientifically, and uses the ratio of GDP to the sum of sulfur dioxide and smoke emissions. The higher the value, the stronger the intensity of environmental regulation.

In addition to the explanatory variable of ER, it also introduces four control variables: science and technology investment, FDI, fixed asset investment and financial development level. Among them are:Science and technology investment (*TI_it_*): previous studies have shown that there is a correlation between science and technology investment and GTFP. Soete et al. [23] found that, for the Netherlands, for the period 1968–2014, extra investment in public and private R&D had a clear positive effect on TFP growth and GDP. Minford and Meenagh [24] found that subsidies offset the frictional costs associated with R&D, incentivizing innovation and thereby stimulating productivity growth. Thursby and Thursby [25] applied nonparametric programming techniques to survey data from 64 universities to calculate TFP growth in each stage. The results suggest that increased licensing is due primarily to an increased willingness of faculty and administrators to license, and increased business reliance is due to external R&D, rather than a shift in faculty research. Therefore, this paper uses the stock of science and technology expenditure to measure the investment in science and technology.Foreign direct investment (*FDI_it_*): a large number of studies and theories have shown that there is a correlation between investment and GTFP. Wang et al. [26] found agricultural FDI has a significant promoting effect on agricultural GTFP. Li and Tanna [27] provided new empirical evidence on the relationship between inward FDI and TFP growth using cross-country data for 51 developing countries over the period of 1984–2010. Their results suggest a weak direct effect of FDI on TFP growth and a robust FDI-induced productivity growth response, which was dependent on these absorptive capacities. In addition, Hu et al. [28], Fernandes and Paunov [29], Kimura and Kiyota [30], Liu and Wang [31], and Suyanto et al. [32] also proved the correlation between FDI and TFP. In this paper, FDI is measured by the actual use of foreign capital stock.Fixed asset investment (*FI_it_*): fixed asset investment plays an important role in improving intellectual capital and the economic development environment [33,34]. Reasonable investment in fixed assets can reduce the cost of industrial development and avoid environmental degradation caused by unreasonable investment [35]. Green economic development is inseparable from large-scale infrastructure construction. However, the negative effects of environmental pressure and resource mismatch brought about by the construction process are not conducive to the development of a green economy [36,37]. In this paper, the level of fixed asset investment is measured by the total fixed asset investment of the whole society in each city.Financial Development (*FD_it_*): Giang et al. [38] investigated the causal effects of access to finance on the TFP of SMEs operating in the manufacturing sector in Vietnam. Their results indicate that improving financial accessibility could directly enhance a firm’s productivity. Arnold et al. [39] investigated the relationship between the productivity of African manufacturing firms and their access to services inputs. The results showed a significant and positive relationship between firms’ productivity and their service performance. Dekle [40] estimated the impact of dynamic externalities using direct measures of TFP growth at the regional level, and found that, at the one-digit level, significant dynamic externalities exist for the finance, services, and wholesale and retail trade industries in Japan. In this paper, the level of financial development is measured by the balance of deposits of financial institutions at the end of the year (Table 1).

The above data are all from provinces’ statistical yearbooks, and the calendar year ‘China city statistical yearbook’.

### 3.2. Model Setting

The Cobb–Douglas production function [41] describes the relationship between the input and output. After taking the logarithm of both sides of the equation, it becomes a regression equation. The model is extended according to the C−D production function form assumptions in Hulten’s [42] theoretical model:(2)$$Y_{it}=A(ER_{it},TI_{it},FDI_{it},FI_{it},FD_{it})F\left(K_{it}^{\theta },L_{it}^{\vartheta }\right)$$
(0<θ<1, 0<ϑ<1)
Y is the output, A is the green total factor productivity, ER is the environmental regulation, TI is the science and technology input, FDI is the foreign direct investment, FI is the fixed asset investment, FD is the financial development level, K is the capital factor and L is the labour factor. The A(·) in Formula (1) indicates the technological progress function.
(3)$$Y_{it}=A\left(\cdot \right)F\left(K_{it}^{\alpha },L_{it}^{\beta }\right)=A_{i0}ER_{it}^{\alpha_{i}}TI_{it}^{\beta_{i}}FDI_{it}^{\gamma_{i}}FI_{it}^{\delta_{i}}FD_{it}^{\varepsilon_{i}}\cdot F\left(K_{it}^{\theta },L_{it}^{\vartheta }\right)$$
where $A_{i0}$ is the initial level of production technology; α, β, γ, δ, ε are the elasticity coefficients of the variables; i represents the city; and t represents time.

The regression model can be set by simultaneously dividing both sides of Equation (3) by $F\left(K_{it}^{\theta },L_{it}^{\vartheta }\right)$, and then taking the natural logarithm.
(4)$$l_{n}GTFP_{it}=l_{n}A_{i0}+\alpha_{i}l_{n}ER_{it}+\beta_{i}l_{n}TI_{it}+\gamma_{i}l_{n}FDI_{it}+\delta_{i}l_{n}FI_{it}+\varepsilon_{i}l_{n}FD_{it}+\rho_{it}+\sigma_{it}$$
$\rho_{it}$ are the individual fixed effects, and $\sigma_{\mathrm{it}}$ are the random error terms.

GTFP may have dynamic inertia, and therefore the first-order lag term of the explained variable is treated as the explanatory variable. However, this would result in the explanatory variable being endogenous, and thus being contrary to the assumption that the explanatory variable is not related to the perturbation term. Therefore, the analysis of the dynamic panel data using traditional OLS methods or fixed effects models for this model will yield biased and inconsistent estimates. The systematic GMM proposed by Blundell and Bond [43] can overcome these problems. The systematic GMM is an extension of the differential GMM method, linking the differential and horizontal equations, using the lag order of the variables as the instrumental variable of the differential GMM method. The lag term of the differential variable as the instrumental variable of the horizontal equation effectively solved the indigenousness problem of the model and obtained consistent estimation results. The final dynamic panel empirical model was constructed as:(5)$$l_{n}GTFP_{it}=l_{n}A_{i0}+\epsilon_{i}l_{n}GTFP_{it-1}+\alpha_{i}l_{n}ER_{it}+\beta_{i}l_{n}TI_{it}+\gamma_{i}l_{n}FDI_{it}+\delta_{i}l_{n}FI_{it}\;+\varepsilon_{i}l_{n}FD_{it}+\rho_{it}+\sigma_{it}$$

Using the systematic GMM method, adding the horizontal equation also increases the number of moment constraints. An AR test and Hansen test are needed to judge whether the estimated results are consistent and the instrumental variables are effective. In this paper, the results of the first-order AR (1) and second-order AR (2) sequence correlation tests for the differential equations and the Hansen test for the validity of the instrumental variables under heteroscedasticity are given with reference to Roodman’s [44] method. The original assumption of the AR (1) and AR (2) tests is that the residual terms of the differential equations allow for first-order sequence correlation but not for second-order sequence correlation. The original assumption of Hansen’s test was that there is no over-restraint of instrumental variables at the time when the instrumental variables are jointly valid.

### 3.3. Analysis Ideas

Considering the availability of variable data, 177 cities above the prefecture level were selected as research subjects, covering the major cities in most provinces in mainland China. At the same time, the subsample will be empirically tested, taking into account that ER may have a differential impact on GTFP across different cities.

On the one hand, air pollution indicators—including SO_2_, NO_2_, CO, PM2.5 and PM10—have been monitored since 2013 in 74 cities of China. This may lead to more attention being paid to environmental pollution and greater measures being taken to reduce environmental pollution in these cities, which could have an impact on the GTFP. Therefore, we divided the sample cities into two categories: key monitoring cities and non-key monitoring cities. Due to the lack of data, we excluded four cities—Taizhou, Suqian, Lishui and Lhasa—and conducted a subsample research on 70 key monitoring cities and 107 non-key monitoring cities (Figure 2).

On the other hand, with the acceleration of China’s industrialization, the gap in the development level between cities is widening, which will cause great differences in resource and energy consumption, and pollution control methods and efforts. Therefore, according to the per capita GDP index of 2016, the sample cities were divided into three categories, according to their economic development levels: high, medium and low. The high economic development level group includes 59 cities, such as Shenzhen and Dongying, and its range of per capita GDP was from ¥66,893 to ¥167,411; the medium economic development level group includes 59 cities, such as Xuzhou and Xiangtan, and its range of per capita GDP was from ¥44,936 to ¥66,845; the low economic development level group includes 59 cities, such as Benxi and Jiujiang, and its range of per capita GDP was from ¥20,987 to ¥44,745.

## 4. Analysis of Research Results

This part focuses on the correlation between ER and GTFP, the differences between them in different types of cities and whether there is a non-linear relationship between them, and tests the robustness of the regression results of the model.

From the results of the correlation test of the first-order AR (1) and second-order AR (2) sequences of the six model difference equations in Table 2, it can be seen that the residual term of the difference equation has a first-order sequence autocorrelation, but there is no second order sequence autocorrelation, which shows that the model setting is reasonable. The *p* values of the Hansen test for the validity of the tool variables are all greater than 0.1, which accepts the original assumption that the tool variables do not have excessive constraints, so the tool variables are jointly valid.

The regression model contains different explanatory variables, some of which belong to the same type of variables. At the same time, the quadratic terms of the ER explanatory variables are added to consider nonlinear effects, and multicollinearity may exist. By investigating the variance inflation factor (VIF), we can find the problem of the multicollinearity-. According to the test results in Table 3, only the $\ln ER_{it}$ and ${\left(\ln ER_{it}\right)}^{2}$ variables had a larger VIF, while the variance inflation factors of other variables were all less than 10, indicating that there was no multicollinearity problem in the system. After the addition of the quadratic term, it is helpful to reduce the deviation of the omitted variables. Although the variance of the estimator increases and the regression coefficient decreases, it is still significant. At the same time, considering that the application of the panel data can better avoid the multicollinearity problem, the F statistics in each model are relatively significant, and the regression results (after adding quadratic terms) can better explain the real problems, while the multicollinearity problem in the model has little impact on the conclusion of the analysis.

### 4.1. Correlation Analysis

According to the GMM method, the model is estimated by gradually adding variables. The results of the model regression (Table 4) show that ${\mathrm{LnER}}_{\mathrm{it}}$ has passed the significance test in the model with different control variables, and is negatively correlated with them, indicating that increasing ER during the research period will reduce the GTFP. The regression coefficient of $\ln GTFP_{it-1}$ is significantly positive, and GTFP has great dynamic inertia and sustainability. The level of GTFP in the previous period will have an impact on the development of the current period. The addition of control variables in each regression model did not change the significance and influence direction of $\ln ER_{it}$. The selection of the control variables was reasonable, and each variable passed the significance test.$\;\ln TI_{it}$ and $\ln FD_{it}$ have a positive influence, while $\ln FDI_{it}$ and $\ln FI_{it}$ have a negative influence. Our results support the views of Hamamoto [10], Soete et al. [23], Minford and Meegagh [24] and Arnold et al. [39]. They also supports the pollution paradise hypothesis in China’s rapid industrialization process. However, the research conclusions of fixed asset investment are not consistent with the views of Antonietti and Marzucchi [36], but are consistent with those of Daina et al. [37] and Giang et al. [38]. More specifically, fixed asset investment does not necessarily promote the development of a green economy and green total factor productivity.

### 4.2. Analysis of Differences among Different Cities

Considering that ER may have different effects on the GTFP of different types of cities, further empirical tests were carried out on the samples. The regression results show that (Table A1) $\mathrm{Ln}{\mathrm{GTFP}}_{\mathrm{it}-1}$ is significantly positive, which indicates that GTFP has great dynamic inertia and sustainability, and the level of GTFP in the previous period will have an impact on the development of the current period. ER passed the significance test in different sample groups, but the direction of action is different. The negative correlation between the key monitoring cities and the non-key monitoring cities shows that strengthening ER cannot promote the growth of GTFP. One possible explanation is that in order to reduce pollutant emissions during the research period, enterprises used the funds that should have been invested in innovative activities to purchase polluting equipment, and environmental investment increased the cost of enterprises, which caused enterprises to reduce their operating efficiency due to the reduction of profits under the condition of constant demand. At the same time, the lack of investment in innovative activities also led to slow technological progress in enterprises, inhibiting GTFP growth. Due to the dynamic inertia and sustainability of GTFP, there is a vicious circle between ER and GTFP. Therefore, in general, the sample cities in the research period are in a state of high investment and low return. In this context, some enterprises—in order to save costs—engage in the phenomenon of the surreptitious discharge of pollutants, which also inhibits the growth of GTFP.

From the regression results of the different development levels of the cities, the ER of cities with a high economic development level is positively correlated with GTFP, while the ER of cities with a medium economic development level and a low economic development level is negatively correlated with GTFP. Among them, the negative correlation coefficient of cities with a low economic development level is larger. This difference shows that high economic development level cities are mostly municipalities directly under the central government or regional central cities. As regional economic centers, they are also regional political and cultural centers. Those cities pay more attention to environmental pollution and invest less in heavy pollution industries; high-tech and service-oriented enterprises account for a large proportion of their economy; the difficulty of environmental governance is low; ER is strengthened; and enterprises can effectively change the mode of production and promote the GTFP. Due to the limited financial funds of cities with a low economic development level, the strength of industrial enterprises is weak, and the investment used for the transformation of green production modes of enterprise is relatively small, thus inhibiting the growth of GTFP. At the same time, in the process of China’s rapid industrialization, cities with low economic development levels tend to gather a large number of resource dependent industries, as well as low-intensity industries of capital and technology. These industries have a large pollutant emission intensity, while the government’s environmental governance capacity is relatively limited, which objectively leads to the increase of ER, and on the contrary, reduces the level of GTFP.

In addition to the correlation between ER and GTFP, the control variables—such as science and technology investment, FDI, and financial development level—have also passed the significance test in the sample regression, indicating that these factors also have an impact on GTFP. However, the influence direction of each control variable is not consistent, and the relationship between ER and GTFP is complicated and uncertain because of the diversification of the influencing factors. Considering the difference of the influence direction of ER in different groups, the relationship between ER and GTFP may be non-linear, and the intensity of ER may have a threshold effect on the development of a green economy, so further study should investigate whether ER has nonlinear influence.

### 4.3. Nonlinear Relationship Analysis

There are two opposite views about the relationship between ER and GTFP. One is that increasing ER will improve GTFP, as in the Porter hypothesis. However, neoclassical economists believe that environmental protection policies will increase the cost of private production and reduce the competitiveness of enterprises, thus offsetting the positive effects of environmental protection on society, which may reduce GTFP [9,15]. Around this debate, we are forced to think about whether the differences in research views are related to other factors besides research objects, research methods and research time intervals. For example, is there a non-linear relationship between ER and GTFP? In other words, does ER have a threshold effect on GTFP? In the papers of Kuznets (1955) [2] and Grossman and Krueger (1995) [45], the square term of the explanatory variable was added to explain the nonlinear relationship between the explanatory variable and explained variable. In order to verify this problem, the square term of the intensity of ER is added into the regression model to further investigate whether there is a non-linear relationship between ER and GTFP, and the final dynamic panel model is:(6)$$l_{n}GTFP_{it}=l_{n}A_{i0}+\epsilon_{i}l_{n}GTFP_{it-1}+\alpha_{i}l_{n}ER_{it}+\tau_{i}{\left(l_{n}ER_{it}\right)}^{2}+\beta_{i}l_{n}TI_{it}\;+\gamma_{i}l_{n}FDI_{it}+\delta_{i}l_{n}FI_{it}+\varepsilon_{i}l_{n}FD_{it}+\rho_{it}+\sigma_{it}$$

From the regression results in Table A2, the AR (1) and AR (2) test results show that there is a first-order sequence autocorrelation in the residual term of the difference equation, but there is no second-order sequence autocorrelation, indicating that the model setting is reasonable. The *p* values of the Hansen test are all greater than 0.1, which accepts the original assumption that there is no excessive constraint on the tool variables, so the tool variables are jointly valid.

From the case of the whole sample, the impact coefficient of ER is negative, the regression coefficient of the square term of ER is positive, and all of them passed the significance test, which shows that there is a ‘U’ relationship between ER and GTFP. This result is consistent with Du et al. [14]. In other words, there is an inflection point in the impact of ER on GTFP. According to the calculation, the inflection point value is 3.62, while the mean value of ER is 5.39, which is on the right side of the inflection point (Figure 3), which shows that the improvement of the current ER intensity has an effect on promoting GTFP, and the positive effect of ER appears.

The ER regression coefficients of the key monitoring cities and non-key monitoring cities are both negative, and the regression coefficients of the square terms of ER are both positive, and all of them pass the significance test, indicating that there is also a ‘U’ relationship between ER and GTFP. Among them, the inflection point value of the key monitoring cities is 9.79, while the average value of ER is 5.86, so it is on the left side of the inflection point (Figure 4A). The environmental pollution problems faced by key monitoring cities are relatively serious, and the current cost of ER measures is high, which fails to play a positive role in promoting the GTFP. The inflection point value of non-key monitoring cities is 4.03, and the average value of ERs is 5.08, so it is on the right side of the inflection point (Figure 4B). The environmental pollution problem of non-key monitoring cities is relatively light, and the environmental burden of enterprise production is relatively small. At present, increasing the intensity of ER will encourage enterprises to carry out more innovative activities, and these innovations will increase the productivity of enterprises, thereby offsetting the costs caused by environmental protection and increasing GTFP.

According to the regression results of the cities with different economic development levels, the regression coefficient of ER variables in high economic development level cities is positive, and passes the significance test. The regression coefficient of the square term of ER is positive, but fails to pass the significance test. This shows that there is no ‘U’ relationship between ER and GTFP, and ER can promote GTFP. The regression coefficients of the ER variables of medium and low economic development level cities are negative, the regression coefficients of the square term of ER are positive, and all pass the significance test, indicating that there is a ‘U’ relationship between ER and GTFP in these cities. Among them, the inflection point value of the medium economic development level cities is 4.18, and the average value of ER is 5.24, so it is on the right side of the inflection point (Figure 5A), indicating that increasing ER will increase GTFP. The inflection point value of the cities with a low economic development level is 5.65, while the average value of ER is 5.04, so it is on the left side of the inflection point (Figure 5B). In recent years, cities with low economic development levels have undertaken some industries with high-energy consumption and high pollution, which were transferred from eastern China. At the same time, these cities have relatively weak financial funds and scientific research strength, and they do not pay enough attention to environmental issues. Therefore, the role of ER in promoting the GTFP has not yet appeared.

### 4.4. Robustness Test

In order to test the robustness of the estimation results, this paper selects the differential GMM method of panel data to re-estimate the model on the basis of System GMM estimation. According to the test results shown in Table A3—the empirical analysis of the linear impact of ER on the full sample, and the key monitoring and non-key monitoring cities—the influence direction of core explanatory variables has not changed, all of them have passed the significance test, and the coefficient value has shown little change. In the non-linear impact analysis, ER and its square term pass the significance test, and the direction of influence has not changed, which shows that there is a ‘U’ relationship between ER and GTFP. The influence direction and significance level of other control variables are basically unchanged, indicating that the research conclusion of this paper is reliable. The regression results grouped by urban economic development level are consistent with the research results of the System GMM, and the research results are robust. The specific results will not be listed here.

## 5. Suggestions

Although Porter affirmed that appropriate ER might promote GTFP, the way in which we define ‘appropriate’ is the key to the problem. Therefore, we put forward the following suggestions.

Firstly, implement policies based on city attributes, and avoid ‘one size fits all’ policies. The impact of ER on GTFP is different and non-linear. The key monitoring cities and low economic development level cities have not yet passed the inflection point, and the task of environmental pollution and control is relatively heavy. It is necessary to further determine the intensity of ER scientifically and reasonably. For example, through mastering the business information of enterprises, the pollutant emission requirements of small and medium-sized enterprises that meet the requirements of the industrial development direction in the early stage should be relaxed through legislation, and the emission standards should be raised after the competitiveness is improved. Avoid the sharp increase of environmental regulation intensity, which will lead to the increase of enterprise operating costs and operating difficulties, which is not conducive to the long-term improvement of GTFP. In non-key monitoring, high economic development level and middle economic development level cities, the industrial structure is relatively reasonable, and there are relatively few high pollution, high energy consumption and low added value projects. The impact of ER variables has passed the inflection point, and the development trend of industrial green transformation is good, which can gradually improve the intensity of ER. Measures such as raising the pollutant discharge standards of enterprises and imposing higher fines on those enterprises that exceed the standards force enterprises to accelerate transformation and upgrading.

Secondly, strengthen supervision and promote the implementation of policy measures. At present, there are still some unreasonable aspects in China’s ER policies. For example, some cities have taken measures to limit production and shut down, affecting the normal production of enterprises. Sometimes, environmental supervision may become a mere formality, and there is a problem of pollution emission fraud, especially in some cities with a low development level, which greatly reduces the effect of enterprise emission reduction and environmental governance. A comprehensive real-time environmental pollution detection system should be established. Cities and related enterprises that violate regulatory requirements should be given an early warning in a timely manner, instead of shutting down enterprises that exceed the standards through surprise inspection. According to the characteristics of the cities, we should determine differentiated ER measures and strengthen the implementation of supervision [46]. We should not only leave living space for small enterprises, but also guide large enterprises in these cities to take the responsibility of environmental governance and jointly promote the GTFP. 

Thirdly, implement comprehensive policies to achieve the advantages of policy mixing. R&D and investment in science and technology is an important guarantee for the improvement of environmental quality, and a lasting driving force for the promotion of GTFP. Under the background of the increasing pressure of industrial transformation and upgrading, and limited R&D funds, the government, R&D institutions and investment institutions should form a joint force. The government should stimulate the R&D enthusiasm of enterprises through preferential loans, tax relief, patent protection and other measures. Enterprises should seek financial support from investment institutions through cooperation with R&D institutions; strengthen green technology innovation, introduction, digestion, absorption and re-innovation; and solve environmental and enterprise green development problems through technological innovation. By improving the pollutant quota trading market, pollution prevention and control will be changed from government compulsion to enterprise conscious market behavior, and from the administrative transaction between government and enterprise to the exchange of economic interests. Investment institutions should improve the level of financial services (especially for non-key monitoring cities with high marginal effect and cities with a low economic development level), encourage the transformation of the real economy of financial services, and promote the improvement of GTFP. In addition, it is vital to strengthen the supervision of investment projects in non-key monitoring, medium and low economic development level cities, to improve the quality of investment, and to prevent the phenomenon of ‘pollution paradises’.

## 6. Conclusions

Based on the panel data of 177 cities above prefecture level in China from 2007 to 2016, this paper empirically studied the correlation between ER and GTFP in the process of industrialization in China, and the differences between different types of cities. This paper then further analyzed the non-linear relationship between ER and GTFP, and tested the robustness of the regression results. The research shows that the relationship between ER and GTFP is complex, which is reflected in the differences and nonlinearity between cities with different monitoring levels and different economic development levels. The specific conclusions are as follows:The relationship between ER and GTFP shows linear and nonlinear differences among different urban groups. There is a linear positive correlation between ER and GTFP in cities with a high economic development level; the higher the level of environmental regulation is, the more favorable it is to GTFP. There is a non-linear correlation between ER and GTFP in cities with a full sample, key monitoring, non-key monitoring, a medium economic development level and a low economic development level, thus presenting a ‘U’ type feature.There are significant differences in the inflection point value of the nonlinear relationship and the location of the mean value of ER variables in different urban groups. The inflexion value of the key monitoring cities is the highest, followed by the low economic development level cities, and the mean value of the ER of these two groups does not exceed the inflexion, so the ER in these two groups are negatively related to the GTFP, and ER has a certain inhibitory effect on the improvement of GTFP. The inflection point values of the whole sample, non-key monitoring and medium economic development level cities are relatively low, and the mean values of the three groups’ ER variables have passed the inflection point, indicating that ER has gradually promoted the GTFP. The improvement of the ER level plays a positive role in green economic development.In addition to ER variables, the control variables also have an impact on GTFP. The investment in science and technology and the level of financial development in different city groups mainly have a positive impact on GTFP, while the variables of FDI and fixed asset investment mainly have a negative impact.

At present, there are different views on the relationship between ER and GTFP, which show the complexity of the relationship. This complexity not only shows that GTFP is affected by many factors—such as ER, investment in science and technology, financial services and FDI—but also shows the non-linearity of this relationship and the difference between different types of cities. In the process of industrialization, developing countries often want to achieve the ‘win–win’ of economic growth and environmental improvement. They hope to promote enterprise innovation and productivity by strengthening ER, but sometimes it backfires. The research results are an empirical analysis based on prefecture-level cities in China’s rapid industrialization process. Due to different national conditions and different stages of economic development, there might be different results. Therefore, whether the research results are applicable to other countries and regions requires further empirical analysis. India, Vietnam and other countries in the process of rapid industrialization may have many similarities with China, and the research results may be useful for environmental policy making in these countries. In the future, if the data of these countries are obtained, we will try to conduct a comparative study.

## Figures and Tables

**Figure 1 ijerph-17-05287-f001:**
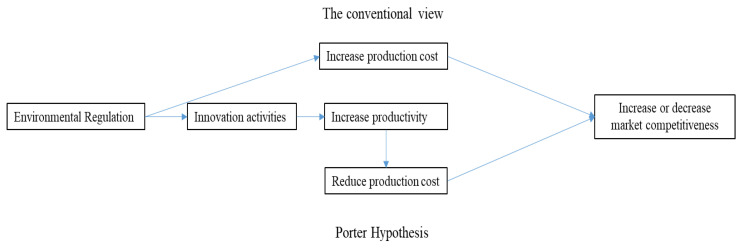
The similarities and differences between the Porter hypothesis and conventional views.

**Figure 2 ijerph-17-05287-f002:**
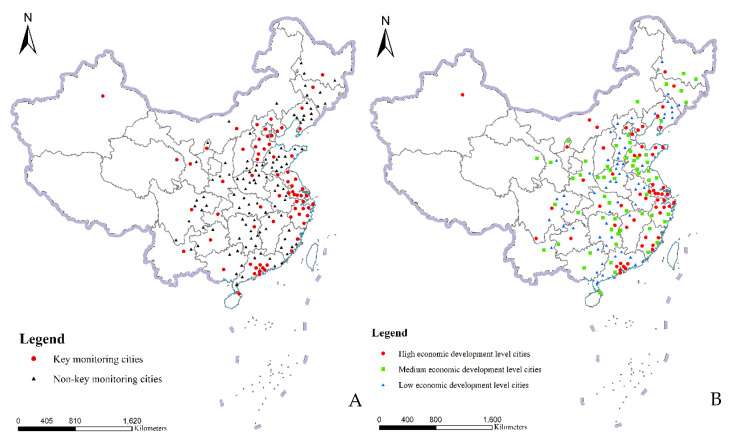
Classification and distribution of research subjects, as grouped by environmental regulation intensity (**A**) or by economic development level (**B**).

**Figure 3 ijerph-17-05287-f003:**
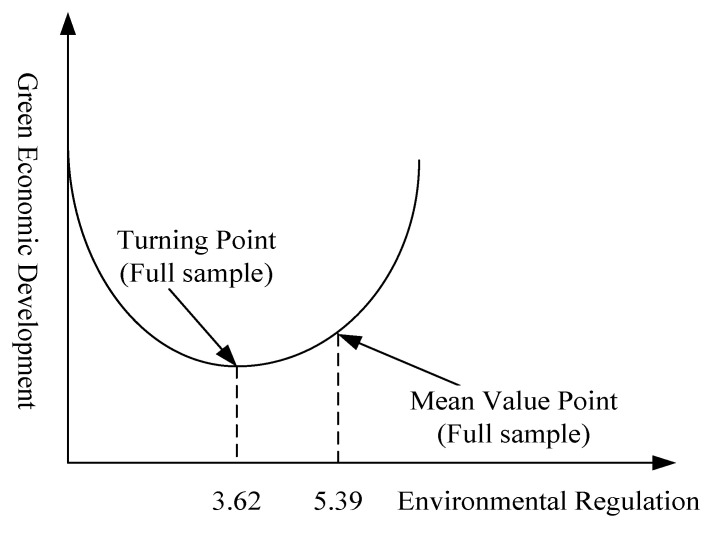
Nonlinear relationship between ER and GTFP (full sample).

**Figure 4 ijerph-17-05287-f004:**
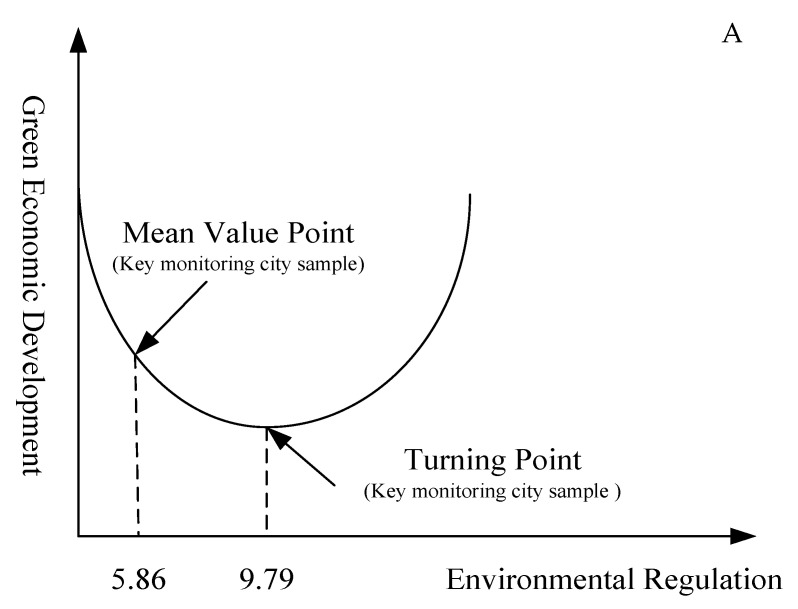
Nonlinear relationship between ER and GTFP. (**A**) Key monitoring cities; (**B**) non-key monitoring cities.

**Figure 5 ijerph-17-05287-f005:**
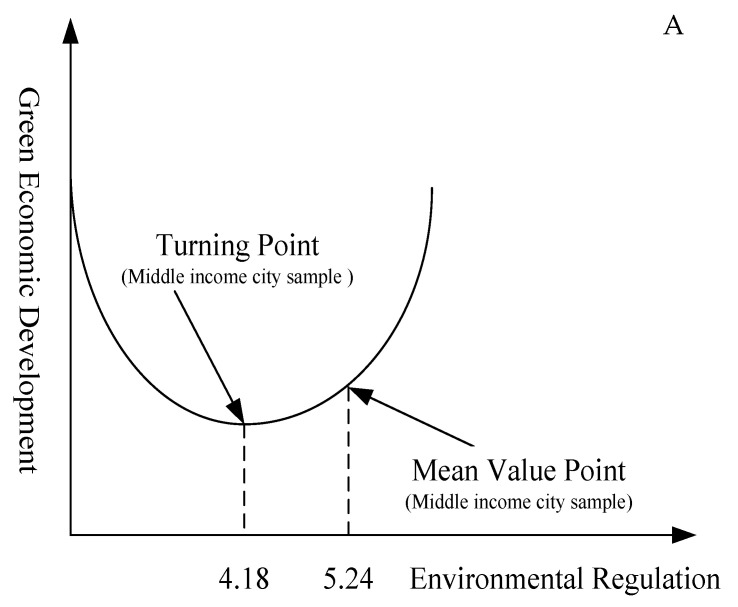
Nonlinear relationship between ER and GTFP. (**A**) Middle-level cities; (**B**) low-level cities.

**Table 1 ijerph-17-05287-t001:** Statistical description of all sample variables.

Variable	Mean	Std. Dev.	Min	Max	Observations
$LnGTFP_{it}$	0.1103	0.2070	−0.7919	1.7990	1770
$LnER_{it}$	5.3902	1.0554	1.3802	10.4219	1770
$LnTI_{it}$	12.1400	2.0758	6.7957	17.5757	1770
$LnFDI_{it}$	11.4600	1.7073	3.9806	15.8742	1770
$LnFI_{it}$	16.1143	0.8796	13.7208	18.9657	1770
$LnFD_{it}$	16.7342	1.0936	13.7708	20.9244	1770

**Table 2 ijerph-17-05287-t002:** System GMM estimation results and the instrumental variable validity test.

Test Type	Full Sample	Key Monitoring City	Non-Key Monitoring City	High-Income City	Medium-Income City	Low-Income City
AR (1)	0.0002	0.0284	0.0026	0.0110	0.0129	0.0463
AR (2)	0.5432	0.2232	0.8804	0.4107	0.3839	0.6285
Hansen	0.441	0.602	0.248	0.253	0.634	0.286

Note: AR (1), AR (2) and the Hansen test report the *p* value corresponding to the statistics.

**Table 3 ijerph-17-05287-t003:** Variance inflation factor (VIF) for each regression group.

Variable	Full Sample	Key Monitoring	Non-Key Monitoring	High-Income	Medium-Income	Low-Income
$\ln ER_{it}$	14.80	11.15	9.91	8.92	10.82	12.44
${\left(\ln ER_{it}\right)}^{2}$	13.36	9.81	9.02	6.71	8.69	10.85
$\ln TI_{it}$	2.27	1.72	1.82	2.31	1.98	1.59
$\ln FDI_{it}$	3.01	3.76	1.89	4.02	2.03	1.75
$\ln FI_{it}$	4.15	4.97	3.47	4.41	3.80	3.16
$\ln FD_{it}$	5.02	4.83	3.76	4.02	3.71	3.47
Mean VIF	7.10	6.04	4.98	5.07	5.17	5.54

**Table 4 ijerph-17-05287-t004:** Full sample regression results (dependent variable: GTFP).

Variable	Model 1	Model 2	Model 3	Model 4	Model 5
$\ln GTFP_{it-1}$	0.9074 ***	0.9149 ***	0.9072 ***	0.9158 ***	0.9112 ***
	0.0180	0.0225	0.0239	0.0251	0.0240
$\ln ER_{it}$	−0.0206 ***	−0.0204 ***	−0.0183 ***	−0.0165 ***	−0.0174 ***
	0.0050	0.0051	0.0051	0.0051	0.0051
$\ln TI_{it}$		0.0028 ***	0.0001 ***	0.0004 **	0.0019 ***
		0.0049	0.0060	0.0064	0.0059
$\ln FDI_{it}$			−0.0102 **	−0.0076 **	−0.0082 *
			0.0050	0.0052	0.0049
$\ln FI_{it}$				−0.0304 ***	−0.0359 ***
				0.0103	0.0114
$\ln FD_{it}$					0.0151 **
					0.0111
CONS	0.2641 ***	0.2996 ***	0.3745 ***	0.8331 ***	0.6588 ***
	0.0327	0.0611	0.0711	0.1725	0.2012
Urban Fixed Effect	Control	Control	Control	Control	Control
Time Fixed Effect	Control	Control	Control	Control	Control

Note: * significance level: 10%, ** significance level: 5%, *** significance level: 1%.

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
