# Peer review of "The Relationship between Environmental Regulation and Green Total Factor Productivity in China: An Empirical Study Based on the Panel Data of 177 Cities"

_ijerph, 2020, doi:10.3390/ijerph17155287_

Round 1
Reviewer 1 Report
Overall, this is an interesting paper providing new insight into the relationship between environmental regulation and green total factor productivity. The paper highlights the complexity of that relationship and demonstrates using data for Chinese cities that it can vary from a positive relationship to a negative one depending on city context.
A few suggestions for improvements:
- The theoretical background is a bit short and it would be relevant to expand that part incl. further discussion of the considered theories and concepts. Here it may be relevant to bring in Banister and Berechman (2000) Transport investment and economic development and Hickman, Givoni, Bonilla & Banister (2017) Handbook on transport and development. For example, it would be good to discuss how economic development and GTFP are connected.
- For the empirical part it would be relevant to discuss further how ER can be measured. One particular, measure is used for the estimation and it could be relevant to consider whether the same results would be obtained with other measures used for ER
- The relationship between the explanatory variables (incl. ER) and GTFP is assumed to be multiplicative. It would be relevant to explained why this functional form was chosen
- As part of the empirical analysis it could also be important to consider the possibility of multicollinearity being present among the explanatory variables.
- A brief discussion of the extent of transferability of the results to other geographical areas may be relevant
Reviewer 2 Report
Thanks for the opportunity to know more about the Environmental Regulation in China's cities. It is exciting.
I consider paper to be an investigation that has been very well accomplished in general terms and fulfilling the basic points of investigation, development and presentation of an article. -The paper is nice and the rationale is good. It's a good econometric application. -Methods
Please the authors use regression analysis. Explain why and justify better. Also what advantage found authors using it instead of other alternative techniques.
-The concluding remarks are quite generic. Highlight implications for the novelty of the study carried out. It states about insightful suggestion and implications. However, I suspect if there is any such specific suggestions or implications included.
-Please see my short comments inside the reviewed file.

Reviewer 3 Report
This is a crucial topic and the paper presents interesting results with useful policy implications. The introduction should be clearer with the objectives and the study significance. If feasible, some of the results should be presented as figures and charts, and complex statistical stuff should be placed in an annex. The results can be better discussed against the literature to flesh out their features and implications in the field. The conclusion should be better aligned with the interpretation of the results and signify sharper policy implications. Ultimately, the paper should be professionally proofread to rectify some grammatical and syntax errors.
Author Response
Please see the attachment.

This manuscript is a resubmission of an earlier submission. The following is a list of the peer review reports and author responses from that submission.